# PCB Defect Detection via Local Detail and Global Dependency Information

**DOI:** 10.3390/s23187755

**Published:** 2023-09-08

**Authors:** Bixian Feng, Jueping Cai

**Affiliations:** 1Xidian University, Xi’an 710126, China; bxfeng@stu.xidian.edu.cn; 2Wide Band Gap Semiconductor Technology State Key Laboratory, Xidian University, Xi’an 710071, China

**Keywords:** defect detection, global dependency, printed circuit board, vision transformer

## Abstract

Due to the impact of the production environment, there may be quality issues on the surface of printed circuit boards (PCBs), which could result in significant economic losses during the application process. As a result, PCB surface defect detection has become an essential step for managing PCB production quality. With the continuous advancement of PCB production technology, defects on PCBs now exhibit characteristics such as small areas and diverse styles. Utilizing global information plays a crucial role in detecting these small and variable defects. To address this challenge, we propose a novel defect detection framework named Defect Detection TRansformer (DDTR), which combines convolutional neural networks (CNNs) and transformer architectures. In the backbone, we employ the Residual Swin Transformer (ResSwinT) to extract both local detail information using ResNet and global dependency information through the Swin Transformer. This approach allows us to capture multi-scale features and enhance feature expression capabilities.In the neck of the network, we introduce spatial and channel multi-head self-attention (SCSA), enabling the network to focus on advantageous features in different dimensions. Moving to the head, we employ multiple cascaded detectors and classifiers to further improve defect detection accuracy. We conducted extensive experiments on the PKU-Market-PCB and DeepPCB datasets. Comparing our proposed DDTR framework with existing common methods, we achieved the highest F1-score and produced the most informative visualization results. Lastly, ablation experiments were performed to demonstrate the feasibility of individual modules within the DDTR framework. These experiments confirmed the effectiveness and contributions of our approach.

## 1. Introduction

With the emergence of Industry 4.0, production processes have been enhanced by incorporating cyber-physical systems that utilize an increased number of circuit boards to create intelligent systems. To ensure the integrity of the circuit board layout, each element needs to be carefully designed, including the through-holes in the hardware, to guarantee high operational reliability. However, due to process uncertainty and noise, ensuring the integrity of all produced circuit boards becomes challenging. Nevertheless, various machine vision-based methods have been introduced to detect defects. With the upgrading of the PCB production process, the circuit density of PCBs is increasing. The defects generated during the PCB production process exhibit characteristics such as small area, large quantity, and different shapes, requiring PCB defect detection methods to be highly precise and fast. The rapid and widespread adoption of deep learning algorithms has led to the development of numerous deep learning-based techniques in the field of electronic circuits, particularly in identifying flaws in printed circuit boards (PCBs). The primary purpose of a PCB is to provide mechanical support for connecting electronic components, achieved through pads, conductive tracks, and soldering. However, environmental factors make PCB surfaces highly vulnerable to quality issues that deviate from design and manufacturing specifications. For instance, Figure 1 displays six types of PCB defects: spur, mouse bite, spurious copper, missing hole, short circuit, and open circuit. These defects not only significantly affect the quality and performance of final products but also result in substantial economic losses for relevant industries. As a result, detecting flaws on PCB surfaces has become a crucial process in managing PCB production quality, attracting significant attention from the industry.

Through the adoption of automated optical inspection (AOI) techniques [1], manual inspections have been largely replaced, leading to enhanced detection accuracy and efficiency. While AOI systems are more convenient and cost-effective than human inspection, they heavily rely on visible imaging sensors, which can be limiting. The quality of PCB images captured by these visible imaging sensors is significantly impacted by illumination conditions, resulting in uneven brightness levels and decreased detection accuracy for various defect types.

Traditional defect detection methods utilize image-processing techniques, prior knowledge, and conventional machine learning approaches to extract low-level features related to defects. However, these methods necessitate the creation of specific classifiers for different defect categories, which restricts their applicability across various application scenarios. In recent years, a range of image processing algorithms have been investigated for PCB defect detection. These include similarity measurement approaches [2], segmentation-based methods [3], and binary morphological image processing [4]. Nevertheless, these techniques require the alignment of inspected images with standard samples during defect inspection. Therefore, there is an urgent need to develop a novel defect detection framework capable of adapting to diverse defect types seamlessly.

The introduction of deep learning has brought significant advancements to object detection, including techniques such as fast R-CNN [5], RetinaNet [6], and You Only Look Once (YOLO) [7], which have demonstrated impressive capabilities in feature extraction. However, when it comes to PCB defect detection, these methods face certain limitations due to the local feature nature of convolutional neural networks (CNN) [8,9]. Defect detection regions on PCBs often occupy only a small portion of the overall image, and even within the same category of surface defects, there can be significant variations in morphology and patterns. While various deep learning-based detectors have been developed to address these challenges, current detectors struggle to simultaneously achieve high detection accuracy, fast detection speed, and low memory consumption. Therefore, there is a need to explore innovative approaches that can effectively address these limitations and meet the requirements of high accuracy, efficient processing, and optimized resource utilization in PCB defect detection.

In recent years, transformer-based deep learning methods have shown remarkable achievements. Within the domain of object detection, transformers [10] have surpassed convolutional neural networks (CNNs) in terms of accuracy. Prominent examples include DETR [11] and Swin Transformer [12]. Unlike CNNs, which are limited to extracting local features within their receptive fields, transformers have the capability to capture global dependency information even in shallow network architectures. This characteristic is especially advantageous for recognition and detection tasks.

However, transformers suffer from the drawback of high computational complexity. To address this issue, a common approach is to divide the input image into patches before feeding them into the transformer. Although this solves the computational challenge, it inevitably results in a loss of local detail information. Therefore, the combination of CNN and transformer has emerged as an optimal solution in numerous tasks across various fields. By utilizing CNN to extract local detail information and transformer to capture global dependency information, superior performance has been demonstrated.

Given the aforementioned problems, we propose a novel PCB surface defeat detection network. To take full advantage of the deep information provided by the source input images, we design a novel two-way cascading feature extractor.

A novel dual cascaded feature extractor, Residual Swin Transformer (ResSwinT), consisting of ResNet and Swin Transformer, is proposed, which can simultaneously focus on local detail information and global dependency information of images. By utilizing the spatial and channel features of spatial multi-head self-attention (SSA) and channel multi-head self-attention (CSA) fusion features, the network can focus on advantageous features. A large number of experiments have been conducted on the PKU Market PCB dataset and DeepPCB dataset, proving that our proposed defect detection converter (DDTR) can better detect difficult defect targets, achieve higher precision defect detection, and improve the yield of PCB production.

## 2. Related Works

### 2.1. PCB Defeat Detection

Over the past few decades, numerous vision-based defect detection methods have been introduced in the field of PCB defect detection. For instance, Tang et al. [13] developed a deep model capable of accurately detecting defects by analyzing a pair of input images—an unblemished template and a tested image. They incorporated a novel group pyramid pooling module to efficiently extract features at various resolutions, which were then merged by groups to predict corresponding scale defects on the PCB. Recognizing the complexity and diversity of PCBs, Ding et al. [14] proposed a lightweight defect detection network based on the fast R-CNN framework. This method leveraged the inherent multi-scale and pyramidal hierarchies of deep convolutional networks to construct feature pyramids, strengthening the relationship between feature maps from different levels and providing low-level structural information for detecting tiny defects. Additionally, they employed online hard example mining during training to mitigate the challenges posed by small datasets and data imbalance. Kim et al. [15] developed an advanced PCB inspection system based on a skip-connected convolutional autoencoder. The deep autoencoder model was trained to reconstruct non-defective images from defect images. By comparing the reconstructed images with the input image, the location of the defect could be identified. In recent years, significant progress has been made in object detection, including the rapid development of algorithms such as YOLO. Liao et al. [16] introduced a cost-efficient PCB surface defect detection system based on the state-of-the-art YOLOv4 framework. Free from the constraints of visible imaging sensors, Li et al. [17] designed a multi-source image acquisition system that simultaneously captured brightness intensity, polarization, and infrared intensity. They then developed a Multi-sensor Lightweight Detection Network that fused polarization information and brightness intensities from the visible and thermal infrared spectra for defect detection on PCBs.

Addressing the challenges posed by small defect targets and limited available samples in the application of deep learning methods to real-world enterprise scenarios for PCB defect detection, this paper presents a novel approach. The proposed method involves a dual-way cascading feature extractor to extract more comprehensive and refined features from PCB images. By employing this feature extractor, the model can effectively capture relevant information for defect detection.

Furthermore, the paper introduces a multi-head spatial and channel self-attention fusion algorithm. This algorithm enables the model to leverage the benefits of focusing on different sizes and channel features of PCB defects. By applying spatial and channel self-attention mechanisms, the model can selectively attend to relevant regions and channels, enhancing its ability to detect defects accurately.

These advancements in feature extraction and attention fusion contribute to overcoming the limitations commonly encountered in PCB defect detection. The proposed approach has the potential to improve the performance and robustness of deep learning models when applied to various enterprise scenarios for PCB defect detection.

### 2.2. Visual Transformer

The Vision Transformer (ViT) architecture, introduced by Google in 2020, has proven to be an effective deep learning approach for a wide range of visual tasks. It serves as a general-purpose backbone for various downstream tasks, including image classification [18], object detection [19], semantic segmentation [20,21], human pose estimation [22], and image fusion [23,24]. Unlike traditional convolutional neural networks (CNNs), ViT eliminates the need for hand-crafted feature extraction and data augmentation, which can be time-consuming. Additionally, ViT can leverage self-supervised learning techniques to train models without labeled data.

In ViT, an image is divided into a grid of patches, and each patch is flattened into a one-dimensional vector. These patch vectors are then processed by a series of Transformer blocks, which operate in parallel and allow the model to attend to different parts of the image. The output of the last Transformer block is fed into a multi-layer perceptron to generate class predictions. ViT has achieved best performance on image classification benchmarks, such as CIFAR, and has outperformed previous methods in multiple computer vision tasks.

Researchers have explored and extended ViT for different applications. For instance, Smriti et al. [25] compared ViT with various CNNs and transformer-based methods for medical image classification tasks, demonstrating that ViT achieved state-of-the-art performance and surpassed CNN and data-efficient Image Transformer-based models. Zhu et al. introduced weakTr [26], a concise and efficient framework based on plain ViT, for weakly supervised semantic segmentation. This approach enabled the generation of high-quality class activation maps and efficient online retraining. Additionally, a saliency-guided vision transformer [27] was proposed for few-shot keypoints detection, incorporating masked self-attention and a morphology learner to constrain attention to foreground regions and adjust the morphology of saliency maps.

In the context of PCB defect detection, the proposed Dual-branch Detection Transformer (DDTR) utilizes a ResSwinT to encode global dependencies and extract comprehensive features. This enables the subsequent detection branch to achieve robust and comprehensive features for defect detection, resulting in notable advancements in detection accuracy for the model.

Overall, ViT has proven to be a versatile and effective architecture in computer vision tasks, and its application and extensions show promising results across various domains, including medical imaging, semantic segmentation, and keypoints detection. In the field of PCB defect detection, the DDTR model leverages the strengths of ViT to improve the accuracy and robustness of the detection process.

## 3. Methodology

Since the introduction of the Swin Transformer, numerous methods employing this architecture have demonstrated remarkable performance in object detection. Given its unique ability for parallel computing and managing global dependencies, the Swin Transformer is employed to extract more comprehensive object information. Furthermore, traditional CNNs can be employed to uncover edge features through shallow convolutional layers and high-level features through deeper layers. This paper proposes their combination to offer abundant semantic information for subsequent detections. Additionally, we introduce a multi-source self-attention fusion strategy to bolster the robustness and flexibility of our model.

### 3.1. Overall Architecture

The structure of our proposed DDTR as shown in Figure 2 is similar to the existing object detection network Cascade R-CNN [28]. It can be divided into a backbone for feature extraction, a neck for feature enhancement, and a head for recognition and detection.

Firstly, the image $X\in R^{H_{0}\times W_{0}\times C_{0}}$ is input into a dual backbone network called ResSwinT composed of Resnet and Swin Transformer. The multi-scale features obtained by ResSwinT are represented as $X_{i}\in R^{H_{i}\times W_{i}\times C_{i}}$, where i=1,2,3,4. In the neck, feature enhancement is performed by mixing convolutional layers and Transformer. Due to the various shapes of defects on PCBs, DDTR introduces a spatial attention mechanism to enable the network to adaptively perceive important spatial features. Furthermore, the features extracted by the backbone exhibit high dimensionality in terms of channels. DDTR will emphasize significant channels through channel attention. Ultimately, the same cascade heads used in Cascade R-CNN are employed to enhance the accuracy of PCB defect recognition and detection in the head of DDTR.

### 3.2. Residual Swin Transformer (ResSwinT)

While traditional single-path CNNs can offer computational and memory efficiency, their extraction of local features alone restricts the model from capturing the broader contextual information present in the input image. This limitation proves critical for the detection of minute defects in PCBs. To address this, we introduce a dual backbone network called ResSwinT, illustrated in Figure 3. ResSwinT combines the residual modules of ResNet with the self-attention mechanism of Swin Transformer, which employs shift windows, to produce multi-scale features encompassing both global and local information within the feature space.

In the initial stage of ResSwinT, the image $X\in R^{H_{0}\times W_{0}\times C_{0}}$ will generate a feature $X_{0}\in R^{H_{0}/4\times W_{0}/4\times 112}$ through the stem layer containing a partition and a convolutional layer. The input image *X* in the partition is divided into patches of size 4×4 and flattened to obtain $X_{p}\in R^{H_{0}/4\times W_{0}/4\times 48}$. The stem layer contains convolution and max-pooling with a stride of 2, and its output is $X_{c}\in R^{H_{0}/4\times W_{0}/4\times 64}$. So the calculation process is
(1)$$\begin{array}{c} X_{p}=Partition\left(X\right) \end{array}$$
(2)$$\begin{array}{c} X_{c}=Maxpool\left(Conv\left(X\right)\right) \end{array}$$
(3)$$\begin{array}{c} X_{0}=X_{p}⊕X_{c} \end{array}$$
where ⊕ represents the channel concatenation.

The subsequent structure of ResSwinT consists of four stages, each consisting of multi-layers perceptron (MLP), a residual part, and a swin part. In order to load the pre-trained weights from ResNet and Swin Transformer, we do not change the structure of the residual and swin parts. The input feature $X_{i-1}$ in i-th stage is first adjusted through MLP to match the channel in the pre-trained network. The residual part of i-th stage contains $n_{i}^{r}$ residual layers, which are composed of convolution, batch normalization (BN) [29], ReLU [30] and shortcut, as shown in Figure 4. Its calculation process is
(4)$$f^{r}\left(X\right)=Res\left(X\right)+X,$$
where Res(·) represents three convolution layers in the residual layer. By using the shortcut of the residual layer, the degradation problem of deep networks can be solved.

Due to partition in the stem layer, only MLP is used to adjust the channel in stage 1. However, in stage 2, 3, and 4, down-sampling is performed through partition before feature extraction by swin transformer blocks. The swin transformer blocks contain two transformer encoders, which are composed of multi-head self-attention (MSA), feed forward (FF) network, and layer normalization (LN) [31], as shown in Figure 4. Unlike the MSA of the transformer, the swin transformer adopts window multi-head self-attention (W-MSA). In the W-MSA of the first encoder, the feature X only calculates local dependency information within the window of (*w*, *w*), as shown in Figure 4. In the next encoder, the window is shifted by (w/2,w/2) to expand the area of extracting dependency information, as shown in Figure 4. Its calculation process is
(5)$$\begin{array}{c} Z_{i}^{t}=W-MSA\left(LN\left(X_{i}^{t}\right)\right)+X_{i}^{t} \end{array}$$
(6)$$\begin{array}{c} {\hat{X}}_{i}^{t}=FF\left(LN\left(Z_{i}^{t}\right)\right)+Z_{i}^{t} \end{array}$$
(7)$$\begin{array}{c} {\hat{Z}}_{i}^{t}=SW-MSA\left(LN\left({\hat{X}}_{i}^{t}\right)\right)+{\hat{X}}_{i}^{t} \end{array}$$
(8)$$\begin{array}{c} X_{i}^{t}=FF\left(LN\left({\hat{Z}}_{i}^{t}\right)\right)+{\hat{Z}}_{i}^{t} \end{array}$$
Through $n_{i}^{s}$ swin transformer blocks in the i-th stage, global dependency information can be gradually extracted with less computational cost.

In summary, the calculation of ResSwinT is
(9)$$\begin{array}{c} X_{i}^{r}=f_{i}^{r}\left(W_{i}X_{i-1}\right) \end{array}$$
(10)$$\begin{array}{c} X_{i}^{s}=f_{i}^{s}\left(W_{i}X_{i-1}\right) \end{array}$$
(11)$$\begin{array}{c} X_{i}=X_{i}^{r}⊕X_{i}^{s} \end{array}$$
where $W_{i}$ is the weight of MLP in i-th stage, $X_{i}^{r}\in R^{H_{i}\times W_{i}\times C_{i}^{r}}$ is the multi-scale features obtained by the residual part $f_{i}^{r}\left(\cdot \right)$, $X_{i}^{s}\in {}^{H_{i}\times W_{i}\times C_{i}^{s}}$ is the multi-scale features extracted by the swin part $f_{i}^{s}\left(\cdot \right)$, where:(12)$$H_{i}=H_{0}/\left(4\times 2^{\left(i-1\right)}\right)$$
(13)$$W_{i}=W_{0}/\left(4\times 2^{\left(i-1\right)}\right)$$
(14)$$C_{i}^{r}=C_{1}^{r}\times 2^{\left(i-1\right)}$$
(15)$$C_{i}^{s}=C_{1}^{s}\times 2^{\left(i-1\right)}$$
where i = 1,2,3,4. Then, $X_{i}\in R^{H_{i}\times W_{i}\times C_{i}}$ generated by the channel concatenation between $X_{i}^{r}$ and $X_{i}^{s}$ is fed into the next stage, where:(16)$$C_{i}=C_{i}^{r}+C_{i}^{s}$$

### 3.3. Multi-Head Spatial and Channel Self-Attention

In the object detection network, the Neck connects the backbone and head, completing the task of feature enhancement. In recent years, multi-scales feature fusion networks have shown significant improvements in accuracy, such as feature pyramid networks (FPN) [32]. We propose a new multi-scale feature fusion strategy named multi-head spatial and channel self-attention (SCSA), as shown in Figure 5. SCSA includes spatial self-attention (SSA) and channel self-attention (CSA), aiming to solve the problem of difficulty in correctly identifying defect targets due to significant differences in PCB defect size, shape, and channel information.

#### 3.3.1. SSA

Due to the large amount of computation involved in the global spatial attention, $X_{i}$ is firstly partitioned according to the size of a×a to obtain $A_{i}\in R^{M_{i}^{A}\times a^{2}\times C_{i}}$, as shown in Figure 6, where $M_{i}^{A}=H_{i}\times W_{i}/a^{2}$. Afterwards, the $A_{i}$ in each region will be clipped into $P_{i}\in R^{M_{i}^{A}\times M_{i}^{P}\times \left(a^{2}\times C_{i}\right)}$ in units of p×p as shown in Figure 5 and the embedded feature ${\hat{P}}_{i}\in R^{M_{i}^{A}\times M_{i}^{P}\times d}$ will be obtained by MLP, where $M_{i}^{P}=a^{2}/p^{2}$. By using an encoder of SSA to extract dependency information within local regions, its structure is shown in Figure 5, and its calculation process can be represented as
(17)$$\begin{array}{c} {\hat{P}}_{i}=W_{i}P_{i} \end{array}$$
(18)$$\begin{array}{c} {\hat{F}}_{i}^{s}=MSA\left(LN\left({\hat{P}}_{i}\right)\right)+{\hat{P}}_{i} \end{array}$$
(19)$$\begin{array}{c} F_{i}^{s}=FF\left(LN\left({\hat{F}}_{i}^{s}\right)\right)+{\hat{F}}_{i}^{s} \end{array}$$
where $W_{i}$ is the embedded weight. The MSA is the same as the MSA in the original transformer. Query vectors $Q\in R^{M_{i}^{A}\times M_{i}^{P}\times d^{\prime }}$ key vectors $K\in R^{M_{i}^{A}\times M_{i}^{P}\times d^{\prime }}$, and value vectors $V\in R^{M_{i}^{A}\times M_{i}^{P}\times d^{\prime }}$ are generated by
(20)$$\left[Q{}_{i},K_{i},V_{i}\right]=\left[W_{i}^{Q}{\hat{P}}_{i},W_{i}^{K}{\hat{P}}_{i},W_{i}^{V}{\hat{P}}_{i}\right],$$
where $W_{i}^{Q}$, $W_{i}^{K}$, and $W_{i}^{V}$ are the weights of the linear layer. Use key vectors to query on the query vectors, and the query results are the sum weights corresponding to the value vectors. The attention calculation process in MSA is as follows:(21)$$\begin{array}{c} MSA(X)=Attention(Q,K,V)\\ =softmax(\frac{QK^{T}}{\sqrt{d^{\prime }}})V, \end{array}$$
where $d^{\prime }$ is the dimension of the vectors. In the calculation process of MSA, all vectors are evenly divided into each head for self-attention.

#### 3.3.2. CSA

The features of the backbone are obtained by concatenating the features of two branches on channel, resulting in a large amount of redundancy in the features. CSA can calculate channel self-attention through spatial embedding encoding, making the network more focused on advantageous channel features.

Similarly, CSA will first partition $X_{i}$ to obtain $A_{i}$, but will not further clip the feature into patches. Secondly, the transformed feature $A_{i}\in R^{M_{i}^{A}\times C_{i}\times a^{2}}$ is used to calculate channel self-attention. By using an encoder of CSA to extract dependency information within local regions, its structure is shown in Figure 5, and its calculation process can be represented as
(22)$$\begin{array}{c} {\hat{A}}_{i}=W_{i}A_{i}^{T} \end{array}$$
(23)$$\begin{array}{c} {\hat{F}}_{i}^{c}=MSA\left(LN\left({\hat{A}}_{i}\right)\right)+{\hat{A}}_{i} \end{array}$$
(24)$$\begin{array}{c} F_{i}^{c}=FF\left(LN\left({\hat{F}}_{i}^{c}\right)\right)+{\hat{F}}_{i}^{c} \end{array}$$
where $W_{i}$ is the embedded weight. In SSA, patches are embedded in the channel dimension, and spatial self-attention is the weighted sum between all patches. However, CSA is embedded features within the channel, and channel self-attention is the weighted sum between channels.

## 4. Experiment Results

### 4.1. Datasets

In this section, the PKU-Market-PCB [33] dataset and DeepPCB [13] dataset are used to validate the performance of our proposed DDTR model.

#### 4.1.1. PKU-Market-PCB Dataset

There are 693 PCB defect images in the PKU-Market-PCB dataset, with an average shape of 2240×2016. PCB defects include six types: missing hole, short, mouse bite, spur, open circuits, and Spurious copper. The image only contains one defect type, but there may be multiple defect targets. The training set contains a total of 541 images, the test set contains 152 images.

Because of the large size of the image, the hardware cannot directly train and test on the initial images. Therefore, we cropped all images into 512×512 patches. Finally, the training dataset contained 8508 images, while the test set contained 2897 images. More detailed information can be found in Table 1.

#### 4.1.2. DeepPCB Dataset

All images in the DeepPCB dataset were obtained from linear scanning CCD, with a resolution of approximately 48 pixels per 1 millimeter. Then, they are cropped into many sub images of size 640×640 and aligned using template matching technology. In order to avoid illumination interference, images are converted to binary image after carefully selecting the threshold. The dataset is manually annotated with six common PCB defect types: open, short, mouse bite, spur, copper, and pin hole. The training set contains a total of 1000 images, the test set contains 500 images, and some instance images are shown in Figure 7. In addition, the number of targets in the dataset is shown in Table 2.

### 4.2. Evaluation Metrics

First, the confusion matrix between the ground truth and the prediction results of the test set is calculated. When the predicted category is the same as the ground truth category, and the Intersection over Union (IoU) between the predicted box and the ground truth box is not lower than the threshold, the prediction is considered correct. True positive (TP) is the number of positive samples for both the ground truth and the predicted result. False positive (FP) is defined as the number of samples with negative ground truth and positive predicted results. True negative (TN) is the number of negative samples for both ground truth and predicted results. False negative (FN) is defined as the number of samples with positive ground truth and negative predicted results.

We used F1-score, which is commonly used in the field of object detection, as the metric for verifying performance. The definition of F1-score is as follows
(25)$$F1-score=\frac{2\times P\times R}{P+R}$$
where P is the precision, defined as
(26)$$P=\frac{TP}{TP+FP}.$$
R is the recall, and the calculation formula is
(27)$$R=\frac{TP}{TP+FN}.$$

### 4.3. Implemental Details

We have designed two types of ResSwinT for DDTR. One is the ResSwinT-T based on ResNet50 and SwinT-T, which has a slightly smaller computational complexity. The another is ResSwinT-S based on ResNet101 and SwinT-S, which has a slightly higher computational complexity. The information for the two types of ResSwinT is shown in Table 3. For SSA in SCSA, the area is 4×4, the patch is 1×1, and the input feature dimension for each head is 32. For CSA in SCSA, the area is 10×10, and the input feature dimension for each head is 25.

To verify the effectiveness of our proposed DDTR, we compared it with six advanced object detection methods, including: (1) one-stage methods: YOLOv3, SSD [34], ID-YOLO [35] and LightNet [36]; (2) two-stage methods: faster R-CNN [37] and cascade R-CNN. During the training and testing process, all methods use a fixed input size of 640×640. All the methods are trained on a Ubuntu18.04 server equipped with E5 2697v3 and RTX3090. Python is 3.7, PyTorch is 1.13.1, and CUDA is 11.7.

### 4.4. Experimental Results

#### 4.4.1. Experimental Results of PKU-Market-PCB

The precision results on the PKU-Market-PCB dataset are shown in Table 4. From the results, it can be seen that the accuracy of all two-stage methods exceeds one-stage object detection methods. A backbone based on the Transformer architecture has higher detection and recognition accuracy compared to CNN. The proposed DDTR method achieved the best results in AP, AR, and F1-scores. Compared to YOLOv3, the DDTR improved 15.42% on F1-score.

From the visualization results in Figure 8, the SSD and YOLO of one-stage object detection have more false alarms. Due to the lack of global dependency information, there are some overlapping target results in CNN-based object detection methods, which are alleviated after using transformer. The proposed DDTR method has good visualization results on the PKU-Market-PCB dataset.

#### 4.4.2. Experimental Results of DeepPCB

The accuracy results of the DeepPCB dataset are shown in Table 5. From the results, it can be seen that the accuracy of all two-stage methods equally exceeds that of the one-stage target detection methods. Compared to CNN, the Transformer-based backbone has higher detection and recognition accuracy. The proposed DDTR method achieved the best results in AP, AR, and F1-score. Compared to YOLOv3, the DDTR has improved 9.04% on F1-score.

From the visualization results in Figure 9, it can be seen that SSD and YOLO have more false alarms. Due to the lack of global dependency information, there are some overlapping target results in CNN-based object detection methods, which have been alleviated by the use of transformer. Due to the lack of attention information, all comparison methods have a significant amount of false positives in the digital area. The proposed DDTR method has good visualization performance on the DeepPCB dataset.

### 4.5. Ablation Experiments

We conducted some ablation experiments on the proposed module, as shown in Table 6, where the best performing ones are highlighted in bold. Firstly, we use ResNet101 Cascade R-CNN as the baseline, which has 0.3813 F1-score on the PKU-MARKET-PCB dataset. If SwinT-S is used to replace ResNet101, it has a 0.89% improvement. If the proposed ResSwinT-S is used as the backbone, it has a 2.89% improvement. On this basis, networks using SSA have a 4.44% improvement compared to Baseline, and networks using CSA is 4.69%. The difference between the SSA and CSA is not significant, indicating that SSA and CSA can enhance the expression ability of features in different dimensions. When ResSwinT-S and SCSA are introduced simultaneously, addition of the feature SSA and CSA shows a 5.99% improvement, while channel concatenation is 6.19%.

## 5. Conclusions

DDTR has designed a new backbone for extracting multi-scale features, named ResSwinT. ResSwinT combines ResNet and Swin Transformer to extract local details and global dependency information. And it can load pre-trained model weights to assist training. Secondly, due to the higher complexity of the features extracted by ResSwinT, we designed a spatial channel multi-head self-attention structure. Spatial multi-head self-attention can encode space features through channel information, and use a self-attention mechanism to achieve weighted summation of spatial features within the region. Channel multi-head self-attention can encode channel features through spatial information, and use a self-attention mechanism to achieve a weighted sum of channel features within the region.

We conducted extensive experiments on the PKU-MARKET-PCB and DeepPCB datasets, and compared to the existing one-stage and two-stage detection models, the proposed DDTR can improve the F1-score by up to 15.42%. The results of multiple visualizations also show that DDTR demonstrates better detection performance. To verify the effectiveness of the module, we conducted a series of ablation experiments. The results of ablation experiments show that ResSwinT and SCSA can improve the accuracy of defect detection.So if DDTR is applied to automated defect detection in the PCB production process, it can accurately detect PCB defects and improve the yield of PCB production.

## Figures and Tables

**Figure 1 sensors-23-07755-f001:**
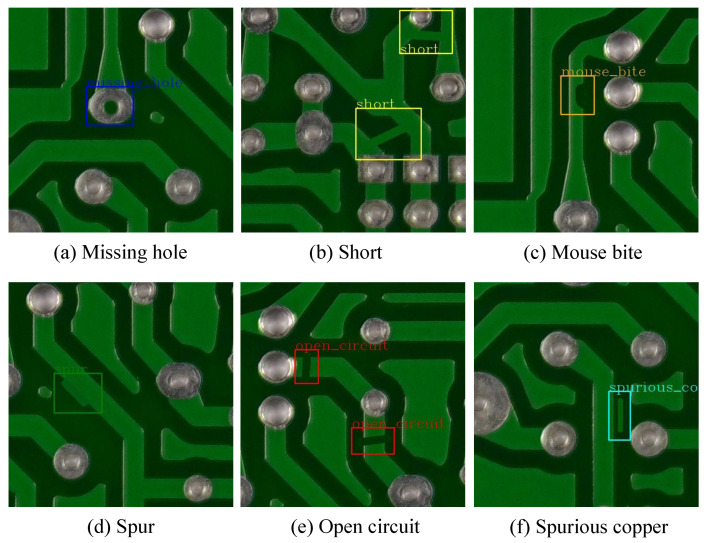
Some defect examples. (**a**) Missing hole, (**b**) Short, (**c**) Mouse bite, (**d**) Spur, (**e**) Open circuit, (**f**) Spurious copper.

**Figure 2 sensors-23-07755-f002:**
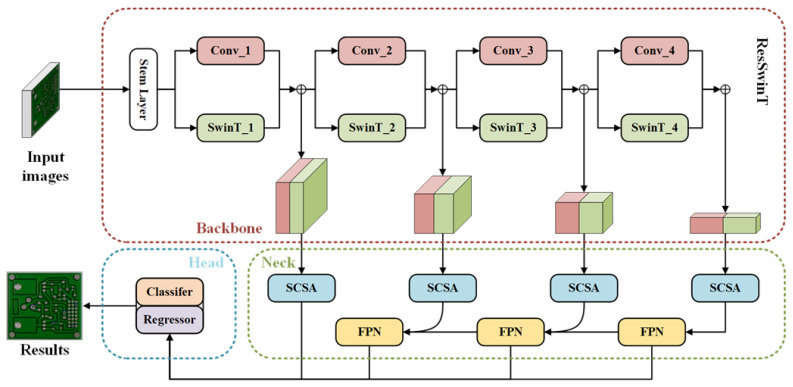
The overall architecture of DDTR. Firstly, the image is input into a dual backbone network called ResSwinT composed of Resnet and Swin Transformer to obtain the multi-scale features. In the neck, feature enhancement is performed by mixing convolutional layers and Transformer. Due to the various shapes of defects on PCBs, DDTR introduces a spatial attention mechanism to enable the network to adaptively perceive important spatial features. Additionally, the features extracted by the backbone exhibit high dimensionality in terms of channels, and DDTR will prioritize crucial channels through channel attention. Lastly, in the head of DDTR, the same cascade heads as those in Cascade R-CNN are employed to enhance the accuracy of PCB defect recognition and detection.

**Figure 3 sensors-23-07755-f003:**
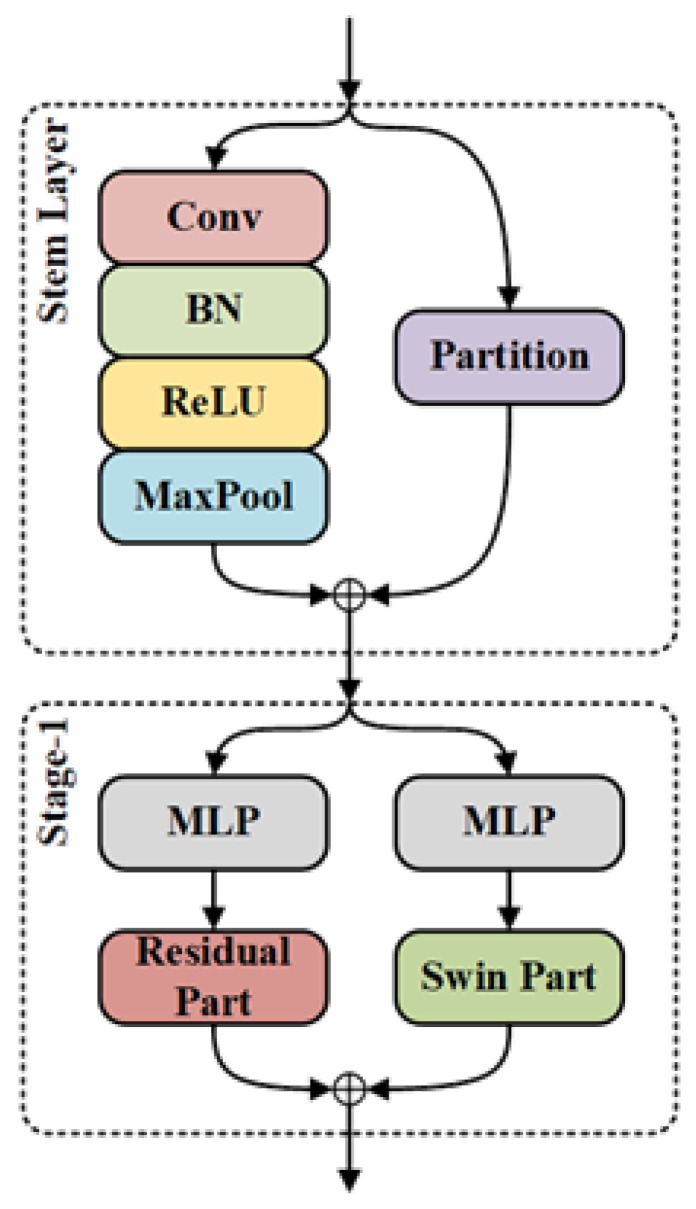
The structure of stem layer and stage-1 in ResSwinT.

**Figure 4 sensors-23-07755-f004:**
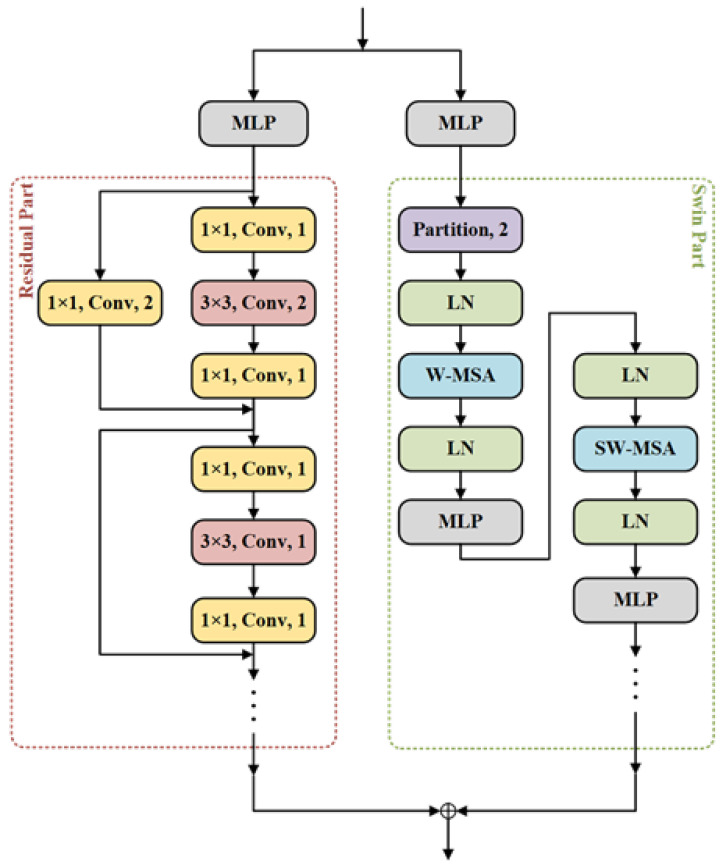
The structure of stage-i in ResSwinT.

**Figure 5 sensors-23-07755-f005:**
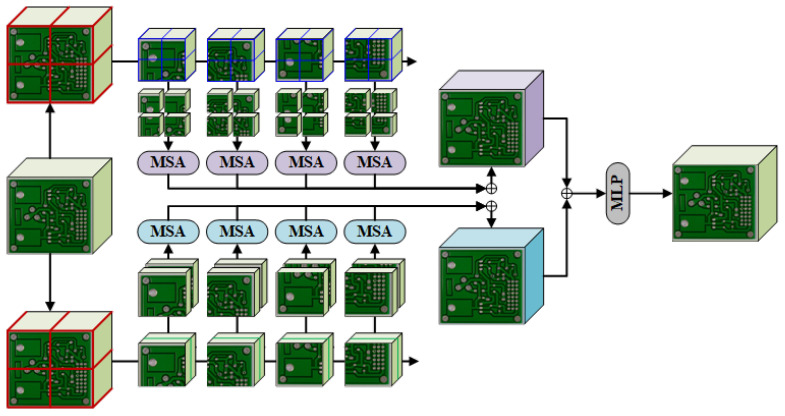
The structure of SCSA. The purple background module is SSA, and the blue background module is CSA.

**Figure 6 sensors-23-07755-f006:**
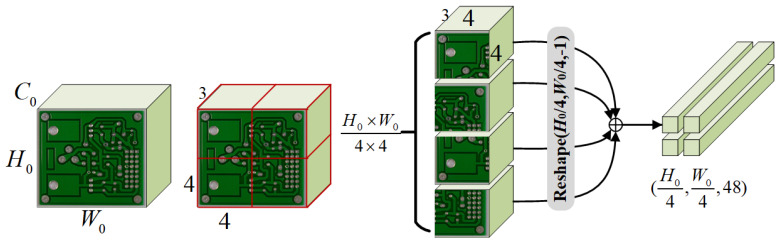
The operation process of partition.

**Figure 7 sensors-23-07755-f007:**
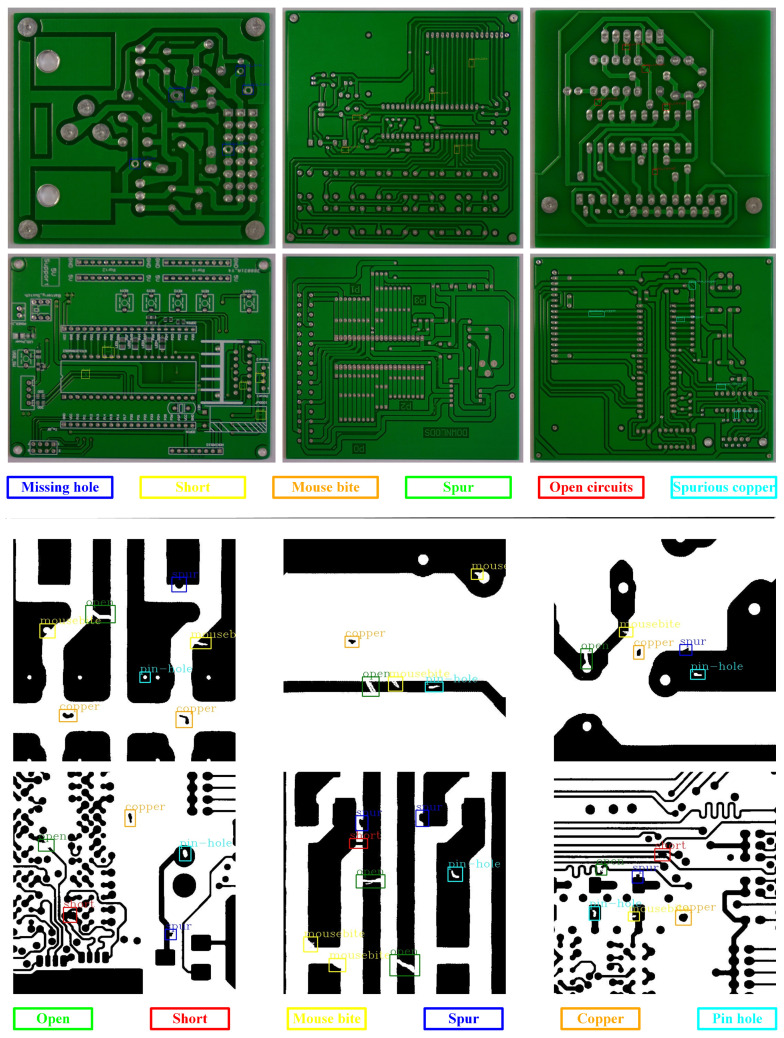
Examples of PKU-Market-PCB datasets and DeepPCB datasets.

**Figure 8 sensors-23-07755-f008:**
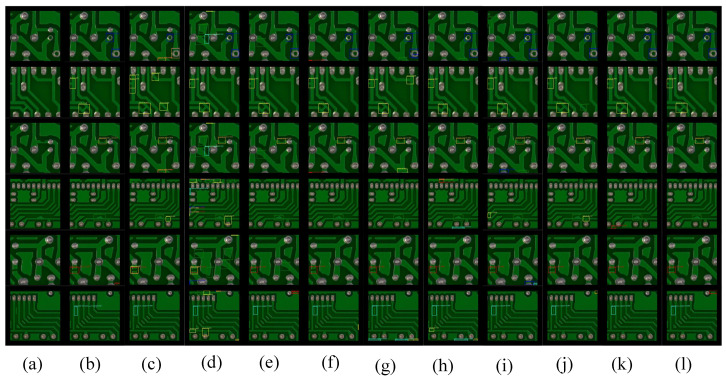
Some visualization results on the PKU-Market-PCB dataset. (**a**) Input image, (**b**) ground truth, (**c**) YOLOv3, (**d**) SSD, (**e**) Faster R-CNN_ResNet50, (**f**) Faster R-CNN_ResNet101, (**g**) Cascade R-CNN_ResNet50, (**h**) Cascade R-CNN_ResNet101, (**i**) Cascade R-CNN_SwinT-T, (**j**) Cascade R-CNN_SwinT-S, (**k**) DDTR_ResSwinT-T, (**l**) DDTR_ResSwinT-S.

**Figure 9 sensors-23-07755-f009:**
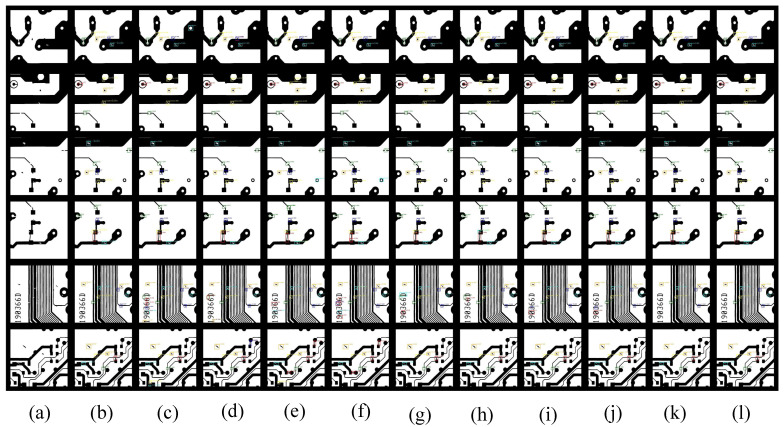
Some visualization results on the DeepPCB dataset. (**a**) Input image, (**b**) ground truth, (**c**) YOLOv3, (**d**) SSD, (**e**) Faster R-CNN_ResNet50, (**f**) Faster R-CNN_ResNet101, (**g**) Cascade R-CNN_ResNet50, (**h**) Cascade R-CNN_ResNet101, (**i**) Cascade R-CNN_SwinT-T, (**j**) Cascade R-CNN_SwinT-S, (**k**) DDTR_ResSwinT-T, (**l**) DDTR_ResSwinT-S.

**Table 1 sensors-23-07755-t001:** The target number of the PKU-Market-PCB dataset.

	Before Cropping		After Cropping	
	Train	Test	Train	Test
Missing hole	362	126	1637	608
Short	351	131	1478	551
Mouse bite	365	126	1661	547
Spur	370	127	1641	587
Open circuits	366	126	1655	587
Spurious copper	371	132	1719	561

**Table 2 sensors-23-07755-t002:** The target number of the DeepPCB dataset.

	Train	Test
Open	1283	659
Short	1028	478
Mouse bite	1379	586
Spur	1142	483
Copper	1010	464
Pin hole	1031	470

**Table 3 sensors-23-07755-t003:** The parameters of ResSwinT.

Layer Name	Output Size	ResSwinT-T		ResSwinT-S	
		Residual Part	Swin Part	Residual Part	Swin Part
Stem layer	160 × 160	Conv, 64, 7 × 7, stride 2 Maxpool, 3 × 3, stride 2	Partition, 4 × 4	Conv, 64, 7 × 7, stride 2 Maxpool, 3 × 3, stride 2	Partition, 4 × 4
		Concatenation, 112		Concatenation, 112	
Stage 1	160 × 160	MLP, 112, 64 $\left[\begin{array}{c} Conv,64,1\times 1,1\\ Conv,64,3\times 3,1\\ Conv,256,1\times 1,1 \end{array}\right]\times 3$	MLP, 112, 48 MLP, 48, 96 $\left[MSA,96,7\times 7,3\right]\times 2$	MLP, 112, 64 $\left[\begin{array}{c} Conv,64,1\times 1,1\\ Conv,64,3\times 3,1\\ Conv,256,1\times 1,1 \end{array}\right]\times 3$	MLP, 112, 48 MLP, 48, 96 $\left[MSA,96,7\times 7,3\right]\times 2$
		Concatenation, 352		Concatenation, 352	
Stage 2	80 × 80	MLP, 352, 256 $\left[\begin{array}{c} Conv,128,1\times 1\\ Conv,128,3\times 3\\ Conv,512,1\times 1 \end{array}\right]\times 4$	MLP, 352, 96 Partition, 2×2 MLP, 384, 192 $\left[MSA,192,7\times 7,6\right]\times 2$	MLP, 352, 256 $\left[\begin{array}{c} Conv,128,1\times 1\\ Conv,128,3\times 3\\ Conv,512,1\times 1 \end{array}\right]\times 4$	MLP, 352, 96 Partition, 2 × 2 MLP, 384, 192 $\left[MSA,192,7\times 7,6\right]\times 2$
		Concatenation, 704		Concatenation, 704	
Stage 3	40 × 40	MLP, 704, 512 $\left[\begin{array}{c} Conv,256,1\times 1\\ Conv,256,3\times 3\\ Conv,1024,1\times 1 \end{array}\right]\times 6$	MLP, 704, 192 Partition, 2 × 2 MLP, 768, 384 $\left[MSA,384,7\times 7,12\right]\times 6$	MLP, 704, 512 $\left[\begin{array}{c} Conv,256,1\times 1\\ Conv,256,3\times 3\\ Conv,1024,1\times 1 \end{array}\right]\times 23$	MLP, 352, 96 Partition, 2 × 2 MLP, 384, 192 $\left[MSA,384,7\times 7,12\right]\times 18$
		Concatenation, 1408		Concatenation, 1408	
Stage 4	20 × 20	MLP, 1408, 1024 $\left[\begin{array}{c} Conv,512,1\times 1\\ Conv,512,3\times 3\\ Conv,2048,1\times 1 \end{array}\right]\times 3$	MLP, 1408, 384 Partition, 2 × 2 MLP, 1536, 768 $\left[MSA,768,7\times 7,24\right]\times 2$	MLP, 1408, 1024 $\left[\begin{array}{c} Conv,512,1\times 1\\ Conv,512,3\times 3\\ Conv,2048,1\times 1 \end{array}\right]\times 3$	MLP, 1408, 384 Partition, 2 × 2 MLP, 1536, 768 $\left[MSA,768,7\times 7,24\right]\times 2$
		Concatenation, 2816		Concatenation, 2816	
Multi-scale outputs		[352 × 160 × 160, 704 × 80 × 80, 1408 × 40 × 40, 2816 × 20 × 20]			

The stride of the first 3 × 3 convolution layer in each stage is 2, and the rest is 1. [Conv, out channel, kernel size], [MLP, in channel, out channel], [Maxpool, kernel size, ], [Partition, area size], [Concatenation, out channel], [MSA, dim, window size, head number].

**Table 4 sensors-23-07755-t004:** The indicator results of various methods on PKU-Market-PCB dataset.

	Metric	Missing-Hole	Short	Mouse-Bite	Spur	Open-Circuits	Spurious-Copper	Average
YOLOv3	AP	0.2303	0.3575	0.2828	0.3362	0.2764	0.3942	0.3129
	AR	0.3512	0.4111	0.4002	0.4157	0.3596	0.4569	0.3991
	F1-score	0.2781	0.3824	0.3314	0.3718	0.3126	0.4232	0.3508
SSD	AP	0.2716	0.3622	0.3045	0.3143	0.3191	0.3415	0.3189
	AR	0.3735	0.4307	0.4066	0.3811	0.4061	0.4414	0.4066
	F1-score	0.3145	0.3935	0.3482	0.3445	0.3574	0.3851	0.3574
Faster R-CNN (ResNet50)	AP	0.3028	0.3521	0.3119	0.2914	0.3331	0.3752	0.3278
	AR	0.3837	0.4285	0.3969	0.3666	0.4155	0.4704	0.4103
	F1-score	0.3385	0.3866	0.3493	0.3247	0.3698	0.4175	0.3644
Faster R-CNN (ResNet101)	AP	0.2958	0.3488	0.3113	0.3166	0.3499	0.3581	0.3301
	AR	0.3840	0.4292	0.4110	0.3789	0.4256	0.4620	0.4151
	F1-score	0.3342	0.3849	0.3542	0.3449	0.3841	0.4035	0.3678
Cascade R-CNN (ResNet50)	AP	0.2873	0.3452	0.3475	0.3153	0.3457	0.3844	0.3375
	AR	0.3908	0.4200	0.4099	0.3779	0.4210	0.4677	0.4145
	F1-score	0.3311	0.3789	0.3761	0.3437	0.3796	0.4220	0.3721
Cascade R-CNN (ResNet101)	AP	0.3152	0.3642	0.3522	0.3074	0.3472	0.3810	0.3445
	AR	0.4061	0.4425	0.4152	0.3889	0.4341	0.4740	0.4268
	F1-score	0.3549	0.3995	0.3811	0.3434	0.3858	0.4224	0.3813
Cascade R-CNN (SwinT-T)	AP	0.3089	0.3605	0.3614	0.3172	0.3431	0.3803	0.3452
	AR	0.3998	0.4358	0.4247	0.3956	0.4218	0.4756	0.4255
	F1-score	0.3486	0.3946	0.3905	0.3521	0.3784	0.4226	0.3812
Cascade R-CNN (SwinT-S)	AP	0.3094	0.3738	0.3195	0.2988	0.3814	0.4079	0.3485
	AR	0.4095	0.4401	0.4026	0.3833	0.4521	0.4879	0.4293
	F1-score	0.3525	0.4042	0.3562	0.3358	0.4138	0.4443	0.3847
ID-YOLO	AP	0.2783	0.3035	0.2960	0.2570	0.3289	0.3504	0.3024
	AR	0.3124	0.3821	0.3501	0.3094	0.3972	0.3975	0.3581
	F1-score	0.2944	0.3383	0.3208	0.2808	0.3598	0.3725	0.3279
LightNet	AP	0.2984	0.3417	0.3155	0.3141	0.3390	0.3788	0.3312
	AR	0.3731	0.4409	0.4136	0.3805	0.4145	0.4813	0.4173
	F1-score	0.3316	0.3850	0.3579	0.3441	0.3729	0.4239	0.3693
DDTR (ours) (ResSwinT-T)	AP	0.3252	0.3742	0.3622	0.3174	0.3572	0.3910	0.3545
	AR	0.4161	0.4525	0.4252	0.3989	0.4441	0.4840	0.4368
	F1-score	0.3651	0.4096	0.3912	0.3535	0.3959	0.4325	0.3914
DDTR (ours) (ResSwinT-S)	AP	0.3294	0.3938	0.3395	0.3188	0.4014	0.4279	0.3685
	AR	0.4295	0.4601	0.4226	0.4033	0.4721	0.5079	0.4493
	F1-score	**0.3729**	**0.4244**	**0.3765**	**0.3561**	**0.4339**	**0.4645**	**0.4049**

AP:AP@0.5:0.05:0.95, AR:AR@0.5:0.05:0.95, F1 = 2 × AP × AR/(AP + AR).

**Table 5 sensors-23-07755-t005:** The indicator results of various methods on DEEPPCB dataset.

	Metric	Missing-Hole	Short	Mouse-Bite	Spur	Open-Circuits	Spurious-Copper	Average
YOLOv3	AP	0.6512	0.6099	0.7158	0.6948	0.8391	0.7319	0.7071
	AR	0.7290	0.6872	0.7802	0.7580	0.8987	0.8628	0.7860
	F1-score	0.6879	0.6462	0.7466	0.7250	0.8679	0.7920	0.7445
SSD	AP	0.6403	0.5598	0.7299	0.7036	0.8737	0.8435	0.7251
	AR	0.7032	0.6385	0.7780	0.7534	0.9017	0.8870	0.7770
	F1-score	0.6703	0.5966	0.7532	0.7276	0.8875	0.8647	0.7502
Faster R-CNN (ResNet50)	AP	0.6426	0.5947	0.7393	0.7011	0.8539	0.8156	0.7245
	AR	0.7109	0.6776	0.7932	0.7594	0.8894	0.8621	0.7821
	F1-score	0.6751	0.6335	0.7653	0.7291	0.8713	0.8382	0.7522
Faster R-CNN (ResNet101)	AP	0.6421	0.5764	0.7286	0.6952	0.8768	0.8512	0.7284
	AR	0.7036	0.6475	0.7749	0.7468	0.9069	0.8938	0.7789
	F1-score	0.6715	0.6099	0.7510	0.7200	0.8916	0.8720	0.7528
Cascade R-CNN (ResNet50)	AP	0.6652	0.6056	0.7561	0.7272	0.9218	0.8779	0.7590
	AR	0.7252	0.6810	0.8038	0.7822	0.9455	0.9355	0.8122
	F1-score	0.6939	0.6411	0.7792	0.7537	0.9335	0.9058	0.7847
Cascade R-CNN (ResNet101)	AP	0.6729	0.6135	0.7537	0.7356	0.9265	0.8810	0.7639
	AR	0.7326	0.6845	0.8048	0.7855	0.9517	0.9326	0.8153
	F1-score	0.7015	0.6471	0.7784	0.7597	0.9390	0.9060	0.7887
Cascade R-CNN (SwinT-T)	AP	0.6880	0.6306	0.7658	0.7403	0.9284	0.8811	0.7724
	AR	0.7451	0.6967	0.8169	0.7977	0.9582	0.9500	0.8274
	F1-score	0.7154	0.6620	0.7905	0.7679	0.9431	0.9143	0.7989
Cascade R-CNN (SwinT-S)	AP	0.6791	0.6365	0.7770	0.7462	0.9355	0.8703	0.7741
	AR	0.7480	0.7079	0.8253	0.8004	0.9621	0.9534	0.8328
	F1-score	0.7118	0.6703	0.8004	0.7724	0.9486	0.9100	0.8024
ID-YOLO	AP	0.6138	0.5730	0.7016	0.6862	0.8703	0.8438	0.7148
	AR	0.7078	0.6850	0.7328	0.7477	0.9246	0.8591	0.7762
	F1-score	0.6574	0.6240	0.7168	0.7156	0.8966	0.8514	0.7442
LightNet	AP	0.6742	0.6184	0.7323	0.7313	0.9321	0.8851	0.7622
	AR	0.7304	0.6879	0.7988	0.7993	0.9314	0.9127	0.8101
	F1-score	0.7012	0.6513	0.7641	0.7638	0.9317	0.8987	0.7854
DDTR (ours) (ResSwinT-T)	AP	0.6823	0.6459	0.7776	0.7577	0.9491	0.9120	0.7875
	AR	0.7473	0.7061	0.8247	0.8101	0.9670	0.9564	0.8353
	F1-score	0.7134	0.6746	0.8005	0.7831	0.9580	0.9337	0.8107
DDTR (ours) (ResSwinT-S)	AP	0.6860	0.6475	0.7825	0.7579	0.9524	0.8911	0.7862
	AR	0.7490	0.7157	0.8343	0.8104	0.9698	0.9551	0.8390
	F1-score	**0.7161**	**0.6799**	**0.8076**	**0.7833**	**0.9610**	**0.9220**	**0.8118**

AP:AP@0.5:0.05:0.95, AR:AR@0.5:0.05:0.95, F1 = 2 × AP × AR/(AP + AR).

**Table 6 sensors-23-07755-t006:** Results of ablation experiment on PKU-Market-PCB.

Cascade R-CNN	AP	AR	F1-Score
ResNet101 (baseline)	0.3445	0.4268	0.3813 (±0.00%)
SwinT-S	0.3485	0.4293	0.3847 (+0.89%)
ResSwinT-S	0.3573	0.4343	0.3921 (+2.82%)
ResSwinT-S/SSA	0.3615	0.4433	0.3982 (+4.44%)
ResSwinT-S/CSA	0.3616	0.4455	0.3992 (+4.69%)
ResSwinT-S/SSA + CSA	0.3668	**0.4500**	0.4041 (+5.99%)
ResSwinT-S/SSA⊕CSA	**0.3685**	0.4493	**0.4049 (+6.19%)**

## Data Availability

The DeepPCB dataset used in this article is available from https://github.com/tangsanli5201/DeepPCB (accessed on 1 July 2023). The PKU-Market-PCB dataset used in this article is available from https://robotics.pkusz.edu.cn/resources/dataset/ (accessed on 1 July 2023).

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
