# Peer review of "PCB Defect Detection via Local Detail and Global Dependency Information"

_sensors, 2023, doi:10.3390/s23187755_

Round 1

Reviewer 1 Report

The manuscript proposed DDTR, which may detect PCB defects via local detail and global dependency information. Some comments are as follows:

 1. Is “DDTR” suitable directly put in the title of the manuscript?

 2. How can the reader see the so-called bidirectional cascaded feature extractor?

 3. Line 105  “[15] developed an ...” looks strange.

 4. Line 179-182

Despite traditional single-way CNN can save a lot of computation and memory, the local features extracted by it make it impossible for the model to obtain the global dependency information of the input image, which is fatal for the detection of small defects in PCBs.

 Do the authors have any proof?

5. Line 187 where does “112” come from?

 6. In Figure 3, stage 1 shows two MLPs. Are they the same? If yes, the process flow should be revised. If not, their names should be revised.

 7. Lines 198 and 208 depict Figure 5, but Line 231 depicts Figure 4. Is this suitable?

 8. Equation (2) is too simple to understand the so-called “calculation process”.

 9. The authors are suggested to re-organized the sentence from lines 214 to 218 to prevent equations inside the sentence.

 10. Preparing a data set for DDTR is tedious because only one kind of defect is allowed. Isn’t it? Do the authors need to prepare a label for each defect in Figure 7?

 11. the authors are suggested to depict the details of DDTR in photos beside Table 3 in order to understand the proposed scheme better.

 12. Line 302 says 8 advanced object detection methods.

 13. The resolution of Figure 8 is too bad to see the annotations after detection. Figure 9 has the same problem.

 14. Figure 9 is wrong.

 15. Table 6 is listed without any explanation.

 16. Why are the results in Table 4 not good for all methods? The AP, AR, F1-score are all below 0.5. The authors are suggested to compare their results with similar research using the same data set.

Author Response

Dear Editors and Reviewers:

        Thanks very much for your valuable comments on our paper. DDTR: PCB defect detection via local detail and global dependency information. In this revision, we have revised the paper: sensors-2531109 " DDTR: PCB defect detection via local detail and global dependency information " according to the reviewers’ comments and made a point-by-point response to these comments. Below are the changes we have made in our revised paper as well as responses to your comments. Thanks a lot for your consideration.

Besides these revisions, we checked the paper carefully and corrected some cryptic and wrong statements to the best of our ability. Moreover, we also checked the text and equations carefully to make them clear. We marked the modified parts in red in the revised marked paper. We hope the paper could have gained in quality this time. Thanks very much for your consideration.

Finally, we are very thankful to the anonymous reviewer for his or her very useful suggestions and comments. We hope the paper could have made an improvement this time.

Best regards to you and your family.

Yours sincerely

. . 2023 .8.10

Reviewer 2 Report

Please see the attachment for detailed comments.

The authors claim for a novel defect detection framework called Defect Detection Transformer (DDTR) is proposed that combines convolutional neural network (CNN) and transformer. Residual Swin Transformer (ResSwinT) is used in the backbone to extract local detail information through ResNet and global dependency information through Swin transformer, thereby extracting multi-scale features and improving feature expression capabilities. The presented study looks interested, however still there is a lot of concern regarding the current form of the paper to be published in the sensors. Therefore, I am going to recommend this paper for a major review. Please give me a descriptive answer and add the changes to the revised version of the manuscript. Please highlight the changes in the manuscript.

Major comments:

1.       Please revise the revised the abstract and make clear the problem statement, why you are doing this work and what is the importance and output of this research work.

2.       The authors wrote the first paragraph about the smart factory and fault detection and PHM without any citing any previous research work? Please refer to the following research paper related the smart factory and fault detection. https://doi.org/10.1093/jcde/qwac015; 

3.       Can the author tell me about the figure 1? Is this related this research work? If not then please give me the reference and highlight in the paper as well?

4.       The author claims that the existing research work has poor accuracy, low robustness, and cannot adapt to changes in the environment for defect detection. How can the author justify this claim?

5.       Please make the contribution as one paragraph and make it clearer and more concise?

6.       The problem statement is not that well written? Please give a comprehensive case study for the available research work? Make a paragraph in the introduction section before the contribution of the paper? Try to explain the challenges and limitations of the available research work and highlight why this research work is needed???

7.       Please make a clear and net figure in the methodology section which can describe the whole process of this research work?

8.       What are the challenges during the data collection please enlist those?

9.       The already built model are used such as Cascade R-CNN? What is the contribution in this? What is the novelty?

10.    Make a paragraph at the end of the results and discussion section that descript the importance of these results and make a descriptive analysis?

11.    Why the author used 2D data? There are extra preprocessing steps involved in the process? Don’t you think it is time taking? Why not use direct the 1D signals?

12.    Please give the description of data conversion with some figures? And why this the 2D data is used?

13.    It would be great to explain the results for the fault detection in the form of bar graphs? Figures can be used tables are hard to analyze? Besides try to add the accuracy graphs and the confusion matrices? It would be nice to explain via the abovementioned graphs. Please revise it.

14.    Can the author let me know about the testing and training the model? How much data is used for the training and testing? Is the testing data is from the same data set or a new environment? If no then how the author can justify the generalization of this model?

15.    Can this model be generalized for data from an unseen dataset? Please justify?

16.    The plagiarisms rate should be decreased it is already 24% please reduce to 10% and show me the proof?

Minor comments:

1.       The equation number should be used for each individual equation? For instance, eq.1 there are a lot of equations (three equations) use equation number for each throughout the whole manuscript?

Author Response

Dear Editors and Reviewers:

        Thanks very much for your valuable comments on our paper. DDTR: PCB defect detection via local detail and global dependency information. In this revision, we have revised the paper: sensors-2531109 " DDTR: PCB defect detection via local detail and global dependency information " according to the reviewers’ comments and made a point-by-point response to these comments. Below are the changes we have made in our revised paper as well as responses to your comments. Thanks a lot for your consideration.

Reviewer 3 Report

Good work has been presented. In my opinion, the abstract needs to be revised. The abstract section is written qualitatively, and the achievements of the article need to be expressed quantitatively.

Author Response

(The authors gave the same response as above.)

Reviewer 4 Report

1.Figures are very low standard, improve it.

2.Modify Experiments Result.

3.Abstract and Conclusion should be rewritten.

4.English must be modified with native speaker.

5.CNN implication is not clear.

1.Figures are very low standard, improve it.

2.Modify Experiments Result.

3.Abstract and Conclusion should be rewritten.

4.English must be modified with native speaker.

5.CNN implication is not clear.

Author Response

(The authors gave the same response as above.)

Round 2

Reviewer 1 Report

In Fig. 9, the input images in the first column are mispositioned, which will result in the mis-corresponding between the input image and the detection image.

Reviewer 2 Report

The paper is published in its current form

The paper is published in its current form

Author Response

Thank you very much for pointing out this issue. We have revised the abstract according to your request.

Reviewer 4 Report

Can be published.

Author Response

(The authors gave the same response as above.)
